# Discussion of the Trust in Vaccination against COVID-19

**DOI:** 10.3390/vaccines10081214

**Published:** 2022-07-29

**Authors:** Jiangbo Fan, Xi Wang, Shuai Du, Ayan Mao, Haiping Du, Wuqi Qiu

**Affiliations:** Institute of Medical Information, Chinese Academy of Medical Sciences and Peking Union Medical College, Beijing 100020, China; fan.jiangbo@imicams.ac.cn (J.F.); wangxi@imicams.ac.cn (X.W.); dushuai@imicams.ac.cn (S.D.); mao.ayan@imicams.ac.cn (A.M.); du.haiping@imicams.ac.cn (H.D.)

**Keywords:** COVID-19 vaccine, trust, vaccine hesitancy, vaccine confidence

## Abstract

The COVID-19 pandemic has introduced serious challenges to global public health security, and the benefits of vaccination via public health interventions have been recognized as significant. Vaccination is an effective means of preventing and controlling the spread of COVID-19. However, trust is a major factor that influences vaccine hesitancy; thus, the distrust of vaccination has hindered the popularization of COVID-19 vaccines. This paper aims to discuss the main problems and the role of trust in the vaccination against COVID-19 to effectively promote and implement policy to promote the acceptance of COVID-19 vaccines.

## 1. Introduction

Coronavirus Disease 2019, referred to as “COVID-19”, is a deadly disease that has introduced serious challenges to global public health security [1]. The significant benefits of vaccination via public health interventions have been recognized. Vaccines are one of the most effective public health measures to prevent and control the spread of COVID-19 worldwide and not only prevent severe diseases leading to hospitalization or death but are also the safest way to achieve herd immunity [2]. According to a global survey on the potential acceptance of COVID-19 vaccines, among 144 countries and regions, the acceptance rate in 72 countries and regions was higher than 60%, especially in China, which was as high as 90%, and 42 countries and regions had rates between 13% and 59%, with acceptance rates being low in the Middle East and North Africa [3,4]. Vaccine hesitancy (VH) has become a factor in vaccine delay and rejections. Trust is the major factor that counters vaccine hesitancy and influences the acceptance of COVID-19 vaccines [5]. This paper aims to discuss the main problems and the role of trust in COVID-19 vaccination to effectively promote and implement the policies supporting the acceptance of COVID-19 vaccines.

## 2. The Role of Trust in COVID-19 Vaccines

Vaccine hesitancy is a global phenomenon, and the underlying factors are multiple, complex, and vary over time and across countries [6]. Complacency, convenience, and confidence are the three main factors influencing vaccine hesitancy [7]. Complacency means a low awareness of disease risk. Convenience refers to vaccine availability, affordability, geographic accessibility, understanding, and the attractiveness of immunization services. Confidence refers to vaccine safety, efficacy, and the capacity of the healthcare system [7]. Trust issues were cited as the main reason for vaccine hesitancy in all countries surveyed [8]. In the administration of COVID-19 vaccines, vaccine trust involves multiple aspects, including trust in the safety and effectiveness of COVID-19 vaccine products; vaccine providers, such as vaccinators and healthcare professionals; and decision makers in healthcare systems and governments [8,9]. The public’s trust in the healthcare system and the government’s ability to make decisions, the safety and effectiveness of vaccine production, and the administration of vaccinators and management by public health professionals are important factors affecting vaccination. This trusting relationship plays a vital role in vaccination uptake. If there is an information gap between vaccine production, providers, and policy makers, vaccine hesitancy and refusal to vaccinate will increase, and a crisis of confidence and anti-vaccination movements will break out [10]. The diversity of the current social media platforms also provides a large number of public opinion positions for anti-vaccination activities [11,12] which have the potential to significantly impact parents’ decision to vaccinate their children [13] and affect public vaccination rates.

## 3. The Main Problems of COVID-19 Vaccines

### 3.1. The Safety and Effectiveness of COVID-19 Vaccines

The safety and effectiveness of the COVID-19 vaccines have always been the main factors influencing whether citizens are vaccinated [10]. Citizens are significantly more likely to be vaccinated against COVID-19 when the vaccine efficacy and the duration of protection are improved and the incidence of major side effects is decreased [14]. However, if the safety and effectiveness of the vaccine cannot be guaranteed, adverse events will be triggered, a serious vaccine crisis will break out, and the public’s willingness to be vaccinated will be greatly affected [15]. Meanwhile, trust in governments, vaccine manufacturers, and the public health system will decline sharply, leading to citizens distrusting vaccines and being anxious about vaccination [16,17]. Governments, vaccine manufacturers, and healthcare systems have to incur large costs in this scenario to allay public panic and regain public trust [15].

### 3.2. Public Confidence and Trust in Government

A global survey of the potential acceptance of the COVID-19 vaccine shows that there are significant differences in the acceptance of the vaccine in different countries and regions. The vaccination rate in China, South Korea, and other countries is higher than 80%, and some countries are slightly lower. This trust helps support citizens to voluntarily vaccinate. Macroscopically speaking, government authorities play an important role in the construction of individual confidence in vaccination. In countries where citizens have a high level of trust in national science, people tend to have more confidence in vaccination [18]. With the outbreak of the COVID-19 pandemic, the credibility of governments around the world has faced even more severe challenges. In 2020, only 55% of citizens in Organization for Economic Co-operation and Development (OECD) countries expressed trust in their governments [19]. The distrust of authorities is mainly concentrated in European countries and is associated with vaccine hesitancy [20,21]. Therefore, effective communication conducted by the government is necessary to enhance government credibility, reduce vaccine hesitancy, and increase vaccination rates [22].

### 3.3. Trust in Public Health Professionals

Trust in scientists and domestic healthcare professionals and confidence in the WHO help promote individuals’ acceptance of COVID-19 vaccines. Although there have been distrust and negative attitudes towards COVID-19 vaccines from a national perspective, a lack of trust in healthcare professionals and healthcare systems has consistently been associated with vaccine hesitancy [5]. Vaccine providers and healthcare providers addressing the issue of vaccine trust during the vaccination process is of vital importance, and their advice helps improve vaccine hesitancy [23,24]. However, compared with the general population, healthcare professionals are more likely to be hesitant to become vaccinated against COVID-19, mainly due to doubts about the safety and efficacy of the COVID-19 vaccine [25,26,27]. Addressing concerns about the safety and effectiveness of vaccines among medical and public health workers, who are at high risk of epidemic prevention and control can, on the one hand, improve the acceptance of vaccines, and on the other hand, fully expand and influence residents’ vaccination decisions and increase vaccination rates.

### 3.4. Authenticity and Reliability of Social Media

Social media are internet-based application platforms where users can express their opinions, exchange topics, and share ideas. However, the content published by users is not strictly reviewed and managed, and their opinions are more likely to lack authority and validity. In addition, the anonymity of users allows them to express their opinions without limitation [28]. At present, social media have become an important channel for citizens to acquire COVID-19 vaccine information and health information, and internet-based interventions have become an important measure to improve citizens’ attitudes toward vaccines [29]. However, information that is not authentic and reliable on the internet often has dramatically powerful potential, and its spread through the internet greatly increases its harm, causing residents to lose trust in COVID-19 vaccines, exacerbating vaccine hesitancy, and reducing vaccination rates [30,31,32]. With the spread of the COVID-19 pandemic, people tend to use social media more. This is usually followed by misinformation on COVID-19 vaccines. Therefore, how to use social media to strengthen health education to improve citizens’ health literacy and reconstruct citizens’ trust in COVID-19 vaccines is a crucial step.

## 4. Policy Recommendations to Address Trust Issues

### 4.1. Improve the Safety and Effectiveness of COVID-19 Vaccines and Increase the Vaccination Rate

The safety and efficacy of COVID-19 vaccines are fundamental measures to improve public confidence. The rapid development of the COVID-19 vaccine may have heightened public concerns over efficacy, availability, and safety [33,34,35]. Despite the widespread availability of the COVID-19 vaccine, its vaccination rate depends somewhat on public opinion and trust. Hence, the government and various regions need to strengthen the supervision of vaccine safety to avoid large-scale adverse reactions after vaccination. Professional and authoritative persons should hold regular press conferences with official media outlets to provide scientific answers to questions of public concern, such as the protective efficacy and sustainability of vaccines. In addition, social media can provide information on vaccine manufacturers and information on the process of vaccine research and development, as well as the principles of vaccines; thus, health education can be achieved by enhancing the openness and transparency regarding vaccine research and development while dispelling the public’s doubts [35].

### 4.2. Enhance the Credibility of the Government and Improve Public Confidence

The credibility of a country and government and the people’s trust in the authorities have always been key factors in the implementation of any public health measure. In the global COVID-19 disease pandemic, people’s concerns about the virus and unoptimistic expectations for the future intensified various vaccination conspiracy theories and rumors, which led to a decrease in compliance with government guidelines and measures [20]. Hence, the government should correct and refute rumors and false information in a timely manner. Meanwhile, there should be channels for the government and the public to participate and communicate effectively in to improve the public trust and goodwill towards the government by enabling the government to provide answers to public concerns about vaccination. In addition, the government can establish a treatment mechanism for suspected adverse reactions to vaccination to deal with the adverse events of vaccination. With the increase in the vaccination rate, adverse events caused by various factors after vaccination usually arouse people’s concern. Therefore, national governments and national disease prevention and control institutions should establish a monitoring system for adverse vaccination events, such as the vaccine advanced event reporting system (VAERS) in the United States [36], the advantage event following immunization (AEFI) in China [37], and the Aus Vax Safety in Australia [38]. At the same time, information about rare adverse events should be disseminated to providers, vaccine recipients, and the public [39].

### 4.3. Strengthen the Role of Health Care Institutions and Improve the Confidence of Professionals

Healthcare institutions are generally considered to be authoritative and professional. Healthcare professionals act as front-line workers for vaccination and play an important role in enhancing public confidence in vaccines and accepting COVID-19 vaccines when providing advice [40]. Therefore, the first step is to reassure healthcare professionals about the safety and efficacy of vaccination and build confidence in COVID-19 vaccines among healthcare professionals [41]. Meanwhile, healthcare professionals also need to keep learning and master the relevant knowledge on COVID-19 vaccines and the countermeasures of adverse reactions so that they can provide professional answers to the inquiries of the vaccinated population and reduce the public’s hesitation and worry about vaccination [42]. Healthcare institutions and professionals, as the direct contacts of the public for vaccination, are important sources of information in terms of public trust, and healthcare professionals should actively encourage public vaccination, reduce public hesitancy through communication and advocacy, support increased vaccination rates, and enhance public trust [25,41,43].

### 4.4. Regulate the Orientation of Public Media Opinion and Strengthen the Credibility of Official Media

While providing unprecedented communication and convenience for the public, social media have also become a major factor affecting the public’s health. Vaccination distrust is not a new phenomenon. With the spread of the global COVID-19 pandemic, some news about anti-vaccination and some disinformation spreading through social media has weakened the public consensus on vaccination acceptance, posing great challenges to improving vaccination rates [32,44]. Hence, governments and authorities need to release official news timely, and the information must be objective, fair, accurate, and comprehensive. At the same time, health institutions’ official accounts should be established on Facebook, Twitter, WeChat, Weibo, TikTok, and other mainstream information platforms to release authoritative information and improve the credibility of official media [28]. Government social media can promote citizens’ active participation, provide timely information about COVID-19 vaccines through official government media and disseminate them widely on networks and platforms with high user volume, increase the public awareness and recognition of COVID-19 vaccines, and through the active processing of information, have a positive impact on citizens’ participation in government social media and prevent public panic and anxiety [45,46,47]. Social media users also need to rein in their behavior, resist anti-vaccine content, and reduce the spread of misinformation [28]. Perhaps, in this epidemic, the dissemination of information did not play a great role, and the reasons behind vaccine distrust are complex and diverse. It may be due to the lack of adequate information dissemination and the lack of response to the actual needs of the public to some extent. It is also necessary to improve the autonomy of every citizen and help them to not believe or spread rumors. Additionally, it is important to ensure that incorrect information is not reported by official channels.

## 5. Summary

COVID-19 vaccines are a safe, effective, and economical measure to prevent, control, and eliminate the spread of the coronavirus, but there are still various problems in increasing the vaccination rate, and vaccine trust is considered to be the main factor affecting COVID-19 vaccine uptake. Regarding the COVID-19 pandemic, improving the safety and effectiveness of vaccines, improving the credibility of the authorities, supporting the role of healthcare institutions, improving public confidence in healthcare professionals, and countering the misinformation in public media are the main measures necessary to increase trust in vaccination against COVID-19.

## Data Availability

Not applicable.

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
