# Peer review of "Discussion of the Trust in Vaccination against COVID-19"

_vaccines, 2022, doi:10.3390/vaccines10081214_

Round 1

Reviewer 1 Report

Dear Authors,

Thank you very much for the opportunity to review your manuscript. Given that it is a "Comment" article and not a research paper, I understand that it is closer to a literature review in its nature than a research article. However, I do feel that some of the statements included in your manuscript are not as well supported as they could be. Additional citations and some copy-editing could be useful to improve the utility of this manuscript to readers. Please see my detailed comments below:

1. Please provide a general citation for the COVID-19 pandemic at the end of the first sentence of the introduction.

2. On the first page, line 21, you state that vaccines "protect individuals from infection". According to my knowledge, the primary goal and effectiveness of most COVID-19 vaccines is to prevent severe disease leading to hospitalization or death, and less so to prevent infection. Please clarify!

3. On the first page, line 26, you have your citation as [2. In general, in-text citation styles are inconsistent throughout, some with just a number 1, and others in square brackets e.g. [5,6]. Please be consistent.

4. Please have a reference at the end of the sentence on page 1, line 28.

5. Same thing at the end of the first and second sentence of section 2.

6. On page 2, line 48, I believe you wanted to have reference 7, but it just says "[".

7. On page 2, line 86, your reference is "[20".

8. On page 3, line 115-118, you state that the safety and effectiveness of COVID-19 vaccines "still require further verification". I'm not sure which specific vaccines you're referring to, and I guess that's part of the problem. If you refer to e.g. the Biontech-Pfizer or Moderna mRA vaccines, or many others, there have been ample studies completed on their safety and effectiveness, to the point that the US FDA approved them for almost all age groups without an Emergence Use Authorization. Saying otherwise, as you do, is going to give fodder to anti-vaccine propaganda, and used as mis- and dis-information to discredit vaccines, which is surely not your intent. So please update this section with more up-to-date information to avoid that.

9. Your suggestions to enhance health education and increase trust in vaccines are very much along the lines of the "deficit model", assuming that people don't trust vaccines because they are ignorant and don't have the correct information, and if we just get the information to them, they will be enlightened, and will choose to do the right thing. If we learned one thing during the pandemic, it is that this framework does not work well. Please consider other approaches as well.

Please see more here: https://en.wikipedia.org/wiki/Information_deficit_model

10. On page 3, line 138, you state that "a treatment mechanism for suspected adverse reactions to vaccination to keep track of any vaccine that is not absolutely safe and effective". I believe this sentence is also misleading, because no vaccines (in fact, nothing) is absolutely safe and effective. Adverse effects can occur to anything, even to eating a cup of sugar. So please simplify this sentence, perhaps just deleting the last phrase.

 11. On page 4, line 143, you mention the VAERS system where people can report adverse reactions to vaccines. This is only a US system, but your comment seems to apply globally. Can you mention similar systems in other countries, such as in China?

12. The entire section 4.3. on page 4 does not have a single reference. Please provide references to your statements.

13. Your references each have a [J] in them. Is that part of the formatting requirements? What does that mean?

14. For reference 16, the author should be OECD instead of OCED.

Author Response

Thank you very much for your amendment. Please see the attachment for the detailed reply to the amendment.

Reviewer 2 Report

The manuscript submitted here as a commentary addresses the problem of vaccine hesitancy with respect to COVID-19 vaccination. Among other things, the authors intend to discuss the causes and background of the reluctance of the population to vaccinate in different nations. In the further text, individual aspects are then taken up and commented on by the authors. These include the areas of "vaccine safety and efficacy", "trust in governments", "trust in medical professionals" and "social media". The authors then go on to provide recommendations for the future on these individual areas.
The manuscript is not a typical life science paper, but rather a social science discussion paper. Apart from the compilation of 36 literature references, no independent data collection or analysis by the authors is evident. Rather, opinions and assessments, which also include sociopolitical evaluations, are reproduced. From an reviewer's point of view, it seems more than questionable whether this manuscript is really within the scope of the journal Vaccines. In any case, the authors should provide a much more balanced and nuanced description of their observations in different political and societal settings before publishing. The general tenor that more authoritarian and centrally governed political systems are associated with higher rates of vaccination acceptance, and that restrictions on free speech in social media have a positive impact on public information, is in urgent need of revision. At a minimum, the advantages and disadvantages of the different approaches would also need to be discussed in light of general human rights and freedom of speech. In many of the democratically structured nations addressed in this manuscript, such debates in society are taking place. In this context, potentially negative effects on the containment of the COVID-19 pandemic are also openly discussed among the population and in the media.
Only with such additions would this manuscript be suitable for publication as a meaningful contribution to the discussion of the trust in vaccination against COVID-19.
As it stands, I cannot recommend the paper for publication in Vaccines.

Author Response

(The authors gave the same response as above.)

Round 2

Reviewer 1 Report

Dear Authors,

Thank you very much for revising your manuscript. I do appreciate the corrections and clarifications that you made. I also appreciate your answer in the author's response in terms of the deficit model, but I would have wanted that specifically mentioned in the manuscript. I disagree with your assessment that the failure of the deficit model is simply due to a lack of sufficient dissemination of information, and lack of responding to the actual demands of the public. I believe that the situation is much more complicated, but I defer to social scientists with exploring that. However, this is a relevant topic for your paper. 

In terms of specifics, I believe the Aus Vax Safety is a system in Australia, not in Europe as you state on line 143 on page 6 of the revised manuscript.

Finally, I was sad to see that reference 19 still said "OCED" for the author instead of "OECD", despite you stating you corrected that in your cover letter.

Sincerely,

Krisztian Magori

Author Response

Thank you very much for your review of this article.  I have revised the Aus Vax Safety and reference 19. I've added the information to the deficit model.

sincerely

Mr.fan

Reviewer 2 Report

In the new version now presented, the authors have included only minor adjustments to their previous version.
My fundamental reservations about this article, which is more an expression of opinion than a scientific study, therefore remain.
The authors' view, which in my view is one-sided in many places, is nevertheless readily apparent to all readers. In this respect, the manuscript can be seen as a contribution to the pluralistic scientific discourse. Against the background of the freedom of speech and science, which is to be striven for worldwide, the publication can therefore be considered. Nevertheless, I am not sure whether such a decision would be taken in the same way all over the world or whether this manuscript would not be suppressed as "false information" elsewhere.
In any case, there is one more factual error that should be corrected, which was added during the revision: In line 143, "and the Aus Vax Safety in Europe" should be replaced by "and the Aus Vax Safety in Australia".

Author Response

Thank you very much for your review of this article. I respect your personal opinions and thank you for supporting the publication of this article.I have corrected the factual errors in the manuscript.